# The Contribution of Frailty to Participation of Older Adults

**DOI:** 10.3390/ijerph19031616

**Published:** 2022-01-30

**Authors:** Debbie Rand, Shelley A. Sternberg, Reut Gasner Winograd, Zvi Buckman, Netta Bentur

**Affiliations:** 1Department of Occupational Therapy, Sackler Faculty of Medicine, Tel Aviv University, Tel Aviv 6997801, Israel; reutvino@gmail.com (R.G.W.); nettabentur51@gmail.com (N.B.); 2Division of Geriatrics, Maccabi Healthcare Services, Modiin 7178051, Israel; sternb_sh@mac.org.il; 3Ateret Rimonim Nursing Center, Bnei-Brak 5137705, Israel; 4Home Care Department, Maccabi Health Care Services, Rishon L’Zion 7526602, Israel; Bukman_tz@mac.org.il

**Keywords:** daily-living, independence, community-dwelling, frailty syndrome

## Abstract

Background: Participation, which is involvement in life situations, is an important indicator of human health and well-being of older adults. Frailty is known to be related to difficulties in activities of daily living (ADL) but the association with participation restriction has not been sufficiently researched. Therefore, we aimed to (1) to assess the correlations between frailty, ADL, and participation; and (2) to identify the contribution of frailty to explaining the participation restriction of older adults. Methods: A cross-sectional study included home visits to community-dwelling older adults aged 75 and older. The Reintegration to Normal Living Index (RNL-I) assessed participation, PRISMA-7 assessed frailty, and the Functional Independence Measure and IADL questionnaire assessed the basic and instrumental ADL. Cognition, which may explain participation, was also assessed (The Montreal Cognitive Assessment) and demographic information was collected. Results: Older adults (N = 121, 60 women), aged 75 to 91 years (mean (SD)—79.6 (3.1)), were included. Older adults demonstrated full to restricted participation (RNL-I-mean (SD)—78.2 (18.0)/100). Frailty was identified in 39 (32%) older adults (mean (SD) PRISMA-7—2.9 (1.4)/7points). A negative moderate significant correlation was found between participation and frailty (r = −0.634, *p* < 0.001). The variance of participation was significantly explained by frailty, 31.5%, and basic ADL, 5.6% (after controlling for age and cognition); the total model explained 44.6% (F = 23.29, *p* < 0.001). Conclusions: Frailty is significantly associated with participation restriction. Since participation has many health benefits, understanding which factors are associated to participation is central to developing interventions for older adults. These findings may help health professionals in the future develop interventions for maintaining and promoting the participation of older adults.

## 1. Introduction

Being active or engaged in activities at an older age has many health benefits, such as preventing cognitive [1] and physical decline [2], improving physical health [3], improving functioning, increasing well-being [4,5], and lowering mortality rates [3]. According to the International Classification of Functioning, Disability and Health (ICF) model of the WHO, participation is the consequence of interactions between an individual’s health status, personal, and environmental contextual factors [6]. Participation, which means involvement in life situations [6], such as meeting friends, taking care of grandchildren, or going to a concert, is an important indicator of human health and well-being [7]; therefore, it is of paramount importance to maintain the participation of older adults.

Participation of older adults can be restricted due to sudden changes in health, such as from a fall or stroke [8], but can also gradually change due to deterioration in health, cognition, functioning, or income [9]. Participation restriction in older adults is prevalent [10,11,12] and has been found to be associated with older age, poor mobility, reduced balance and confidence, and more depressive symptoms [11,13,14].

Frailty is a clinical syndrome of older adults, which can lead to adverse health outcomes such as disability, hospitalization, institutionalization, and death [15,16]. The interaction between older age, and its associated changes, and chronic diseases leads to decreased physiological reserves, which can result in increased vulnerability to frailty [15,17,18]. Decline in physical function, reduced gait speed, reduced endurance, decreased physical activity, and weight loss are commonly identified as components of frailty [15]. Therefore, frail older adults might experience difficulties in participating in activities that are meaningful to them; however, this has not been sufficiently studied.

Gait speed, for example, one of the physical characteristics of frailty [15], which has been described also as the ‘sixth vital sign’ since it is an indicator of health and functioning in older adults, is related to the level of independence [19]. While older adults with increased gait speed (above 0.8 m/s) have the ability to move around the community and cross streets safely, others with reduced gait speed (below 0.4 m/s) might be housebound. Gait speed has been related to frailty in 85 studies of older adults [20]. Although gait in this recent systematic review was found to be an indicator of health and functioning, the relationship between reduced gait speed as one of the characteristics of frailty and participation has not been assessed in many studies. Slower gait speed was found to be significantly associated with social participation restriction in adults 50 years and older in the US [21], although it was based on two self-report questions regarding the social aspect of participation only.

The physical characteristics of frailty have been researched extensively and found to be related to difficulties in basic and instrumental Activities of Daily Living (ADL) [12,14,22,23]; however, there is limited evidence regarding the relationship between frailty and participation. In other words, it is not clear how frailty characteristics (such as reduced gait speed, reduced endurance, and weight loss), might impact the person’s ability to go on trips, go to the theater, take part in social events or even just leave their home. A high incidence of participation restriction was found in two studies that assessed participation among older adults with frailty who were recruited from day care centers and hospitals [10,24]. In community-dwelling older adults, the selection of meaningful activities (physical, cognitive, social, or other daily activity) [25] and activity diversity were found to be associated with frailty [26]. The importance of participation was demonstrated in a 5-year longitudinal study. In this study, performing healthy lifestyle behaviors (i.e., farming, healthy daily activities, and social participation) was associated with lower odds of becoming frail [27].

As opposed to the extensive research of frailty and daily functioning and/or physical characteristics, there is limited research focusing on frailty and participation (as a health condition, using the ICF framework). Due to the importance of participation, in general, and specifically in the lives of older adults, we decided to carry out this study. The aims of this study were (1) to assess the correlations between frailty, ADL, and participation; and (2) to identify the contribution of frailty to explaining the participation restriction of older adults. We hypothesized that participation will be negatively correlated to frailty and that beyond age, cognition, and ADL, frailty will contribute to explain participation restriction. In the future, these findings may help health professionals develop interventions to maintain and promote the participation of community-dwelling older adults.

## 2. Materials and Methods

A cross-sectional study among community-dwelling older adults was conducted.

### 2.1. Participants

Older adults who met the following criteria were included: age 75 or older; community dwelling; able to hear, understand, and speak the language; able to ambulate independently within their homes (with or without a walking device); do not have a diagnosis of dementia in their medical record. Older adults with significant cognitive impairment (a score of less than 16 points on the MoCA [28]), which might indicate having dementia, were excluded from the study.

### 2.2. Tools

The Reintegration to Normal Living Index (RNL-I) [29] was used to assess participation. It assesses seven domains of participation according to the ICF (WHO), such as community mobility, daily activities, recreational and social activities, family role(s), personal relationships, presentation of self to others, and general coping skills. Each domain is rated by a visual analogue scale ranging from 1—“Does not describe my situation” to 10—“Fully describes my situation”. The adjusted score is calculated as the (Total Score/110) × 100, and ranges from 10–100 points, with a higher score indicating more participation. The RNLI has previously been used with community-dwelling rehabilitation populations [30], mainly with older adults after stroke [31,32]. In addition to the total RNL-I score, we used two sub-scores ‘participation in physical activities’ (maximum score of 40 points) and ‘participation in social events’ (maximum score of 70 points) [33].

PRISMA-7 [34] was used to assess frailty. This simple self-report tool is based on the identification of risk factors for functional decline [34], and is feasible, acceptable, and reliable to assess frailty [35]. It consists of seven dichotomous yes/no questions regarding age, gender, general health, social support, and activities, where one point is scored for every ‘yes’ answer. It has a sensitivity of 0.83 and specificity of 0.83 (positive predictive value = 0.40, negative predictive value = 0.97) [36] and a cut-off score of 3 or higher suggests moderate to severe frailty and the need for further assessment [34]. Since the first question asks the person’s age (age above 85?) and our study population was above 75 years old, we used a cut off score of 4 points and above, which was found to have sensitivity and specificity of 74.4% and 87.4%, respectively, and a recommended cut-off point for this questionnaire [37].

The Functional Independence Measure (FIM) [38], which includes 18 activities, was used to assess basic ADL. Each activity was rated from 1 (totally dependent) to 7 (totally independent). Scores ranged from 18 (totally dependent) to 126 (totally independent). The FIM was administered in an interview format [39].

The IADL questionnaire (IADLq) [40] assesses eight instrumental activities of daily living (IADL) (telephone use, grocery shopping, food preparation, light and heavy housekeeping, laundry, transportation, medication use, and handling finances). Each item has three answer categories (not able, able with support, independent). The total scores range from 0 (dependent) to 23 (fully independent).

The Montreal Cognitive Assessment (MoCA) was used to screen for eligibility in addition for assessing cognition. It is a brief cognitive screening tool with high sensitivity and specificity used to distinguish individuals with mild cognitive impairment from older adults with normal cognition [28]. The following cognitive domains are assessed: attention and concentration, executive functions, memory, language, visuo-constructional skills, conceptual thinking, calculations, and orientation; it has a maximum score of 30 points. In addition, participants were characterized with cognitive decline (16–18 points), mild cognitive impairment (19–25 points), and normal cognition (26 points and above) [28].

Demographic information (such as age, sex, living condition, education, and falls) was also collected.

### 2.3. Statistical Analysis

Descriptive statistics were used to describe the population in terms of demographics, participation, daily functioning, frailty, and cognition. Pearson correlations were used to assess the correlations between participation and cognition, frailty, and independence in basic and instrumental ADL.

To assess the differences in participation (total and sub-scores) between ‘frail’ and ‘non-frail’ groups, Independent Samples *t*-test were used. Multivariate linear regression analysis (Enter method) was used to identify the variables that best explained participation. Variables that were significantly correlated to participation were entered into the model. Since age [13] and cognition (e.g., [41]) are factors known to correlate with participation, they were entered first into the regression model in order to control for these. In addition, only basic ADL was entered into the model since independence in basic and instrumental ADL are highly correlated. Scatter plots of residuals against the model data were inspected, as well as outliers and influential data points, and the variance inflation factor for multicollinearity. All analyses were conducted using SPSS for Windows version 25.0 (SPSS, Inc., Chicago, IL, USA).

## 3. Results

One hundred and fifty-four home visits were done. Subsequently, 31 participants were excluded due to low MoCA scores and two participants asked to stop their participation. The study population included 121 older adults with a mean (SD) age of 79.6 (3.1), ranging from 75 to 91 years, with an equal distribution of men and women. Most participants lived with family members and reported that they do not work or volunteer (see Table 1).

Table 2 presents the scores for the tools used to characterize participation, frailty, cognition, and ADL. Participation varied widely from very restricted to full participation; RNL-I total scores ranged from 14.5 to 100 out of 100 points. The RNL-I sub-scores of the physical activities and social events varied widely as well.

Frailty was identified in 39 (32%) older adults, and the mean (SD) PRISMA-7 score was 2.9 (1.4) points, with scores ranging from 0 (not frail) to 6 (very frail). Cognitive status also varied: 19.8% had cognitive decline, 67.8% mild cognitive impairment, and 12.4% demonstrated normal cognition. Most participants were independent in basic ADL, but independence in instrumental ADL varied considerably, from 8 to 23 points, indicating a dependency to full independence.

Negative, moderately significant correlations were found between participation as well as between the sub-scores of participation: physical activities, social events, and frailty (see Table 3). Older adults who were more frail had lower participation. Participation (total and sub-scores) was also moderately significantly correlated to basic and instrumental ADL (older adults who were more independent in ADL had higher participation). Participation (total RNL-I score) was weakly correlated to cognition (older adults with higher cognition had higher participation), but no significant correlation was found with the physical activities sub-score. The total participation score was not correlated to age or sex, but a weak negative significant correlation was found between participation in the sub-score of social events and age (older age had lower participation in social events; see Table 3).

Due to the wide range in participation scores, we assessed the difference in participation between ‘frail’ (N = 39) and ‘non-frail’ (N = 80) older adults. Significant differences (*p* < 0.001) were found between groups for the RNL-I total score (85.2 (11.6) as opposed to 64.6 (20.2)) as well as for the physical activities (35.1 (4.8) versus 24.5 (8.5)) and social events (58.6 (9.1) versus 45.0 (15.4)), respectively (see Figure 1). As can be seen in Figure 1, the ‘frail’ group is more heterogeneous in terms of participation than the ‘non-frail’ group.

A linear multivariate regression analysis was conducted, in which participation was the dependent variable and cognition, frailty, and BADL were the independent variables. Since participation is related to age, we controlled for age by adding it first into the model. The complete model significantly explained 44.6% of the variance in participation (F = 23.29, *p* < 0.001), with frailty contributing 31.5% and basic ADL contributing an additional 5.6% (Table 4). The variance inflation factors, condition indices, and tolerance were within acceptable values, indicating that multicollinearity was not a concern.

## 4. Discussion

In this study, a negative moderate significant correlation was found between frailty and participation. Frailty contributed 31.5% to the variance in participation. Basic ADL contributed an additional 5.6% to the model, which significantly explained 44.6% of the variance in participation of community-dwelling older adults aged 75 and older.

Since participation is considered an important outcome measure following rehabilitation, the participation of older adults with disabilities has been researched [42]; however, limited research has focused on the participation of community-dwelling older adults [13]. It is encouraging to see that, in the last two years (perhaps due to the COVID-19 pandemic restrictions; e.g., [43]), more researchers have become aware of the importance of the participation of older adults.

Participation, which is involvement in physical and social life situations, is a broad concept with several aspects, such as preferences, enjoyment, and satisfaction from activities [44]; therefore, it is challenging to capture or measure. Social participation, for example, which is one aspect of participation, has been researched and found to be related to a lower probability of frailty (e.g., [45]) and even protective against frailty, as found in a five-wave study in Taiwan [46]. The ‘lifestyle activities’ of 895 older adults, including participation in social (i.e., face-to-face interpersonal communication and activities with friends) or intellectual activities (such as reading or solving puzzles) was found to be less prevalent among frail older adults than among non-frail older adults aged 60 years and over [47]. A large health survey in Japan aimed to assess the association between the selection of meaningful activities and frailty. The meaningful activities of community-dwelling older adults (65 and above) with pre-frailty and frailty (N = 859) [25] were grouped as physical, cognitive, social, and other daily activity. The older adults were also grouped according to their impairment: physical frailty, cognitive frailty, social frailty, none, or all types of frailty. The authors concluded that the selection of meaningful activities may be affected by frailty and cognitive impairment, but since no differences in satisfaction and performance were found, further research is needed.

Participation in our study was assessed using RNL-I, which aims to measure the broad concept of participation, specifically the physio-psycho-social characteristics of living a well-adjusted life [29]. The RNL-I total score of our sample (78.2 (18.0) points) was higher (i.e., more participation) than the sample described by Liu and Ma [33] of pre-frail and frail community-dwelling older adults aged ≥65 (68.3 (19.6) points). However, the mean RNL-I total score of our ‘frail’ group (64.6 (20.2)) and the wide variance is similar to this sample from Hong Kong. This Chinese sample of frail and pre-frail older adults had a higher score for physical activities compared to our ‘frail’ participants but similar scores for participation in social events. The participation (assessed by the RNL-I) of frail older adults in Australia (mean age 84) was very restricted, possibly since all participants were frail and were recruited following hospital discharge or a visit to the emergency department [24].

Participation is related to independence in activities of daily living but goes beyond it, since additional domains such as leisure activities and community life, which are essential for wellbeing and life satisfaction [13], are included. Participation of our sample was moderately significantly correlated to independence in basic and instrumental ADL. In other words, older adults who were more independent had higher participation. Similar findings were found in older adults 3 months [48] and 2 years post hip fracture [49]. Their participation was positively associated with independence in ADL and functional mobility and cognitive impairment was associated with participation restriction (lower RNLI scores) [48].

A recent phenomenological qualitative study explored the life experience of 10 frail older adults [50]. The older adults expressed the importance of performing meaningful activities, despite the physical difficulties they experience. Another recent study [26] assessed the diversity of daily activities (in terms of type, frequency, and evenness of different activities performed during the week) among frail older adults. Frail participants (4% of 658 older adults) had significantly lower activity diversity compared to non-frail participants.

Further research is needed to understand what type of participation restrictions are experienced by older adults with frailty. This information can perhaps help develop recommendations for compensating or overcoming the difficulties rather than giving up activities, especially activities that are considered meaningful to the participants. Encouraging older adults to remain active might prevent the development of frailty.

This study has several limitations. Our sample was relatively small and convenience sampling was used, limiting the generalizability to a larger community-dwelling adult population. Chronic diseases and medications were not collected, and thus were not factored into the model. Since this is a cross-sectional study, we cannot infer cause and effect in terms of frailty and participation. The fact that our sample included older adults over age 75 and that home-visits were conducted, allowing us to reach participants that might not have participated otherwise, strengthens our findings.

## 5. Conclusions

Frailty is moderately correlated to participation restriction in older adults. Since participation has many health benefits, understanding which factors impact participation is central to developing interventions for the older adult population. Future studies could assess the efficacy of occupational therapy interventions in frail older adults to improve participation.

## Figures and Tables

**Figure 1 ijerph-19-01616-f001:**
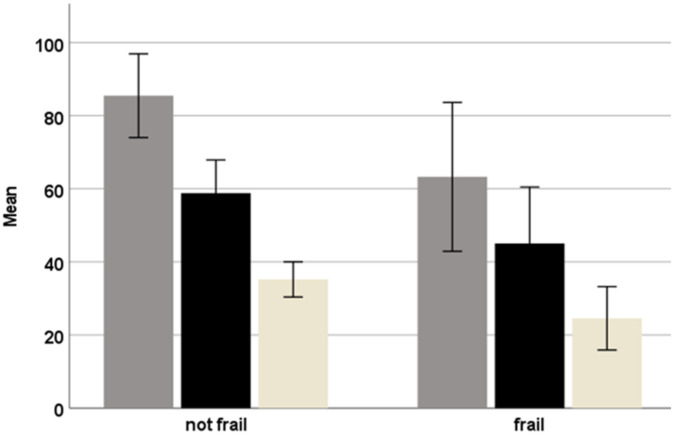
Participation; differences between ‘frail’ (N = 39) and ‘non-frail’ (N = 80) older adults in the mean (SD) RNL-I total (grey) and sub-scores: participation in physical activities (off-white) and participation in social events (black). Please note: the RNLI- total score ranges from 0 to 100 points, participation in physical activities from 4 to 40 points, and participation in social events from 7 to 70 points.

**Table 1 ijerph-19-01616-t001:** Demographic data of the older adults (N = 121).

		N (%)
**Sex**	Male Female	61 (50.4) 60 (49.6)
**Living Arrangement**	Alone With partner/children	32 (26.4) 89 (73.5)
**Education**	0–8 years 9–12 years 13+ years	29 (24.0) 61 (50.4) 31 (25.6)
**Work/Volunteer**	Yes No	24 (19.8) 97 (80.1)

**Table 2 ijerph-19-01616-t002:** Participation, frailty, cognition, and daily functioning of the sample (N = 121).

	Tools	Mean (SD)	Min–Max
**Participation**	RNL-I total (10–100)	78.2 (18.0)	14.5–100
	RNL-I—participation in physical activities (4–40)	31.7 (7.9)	7–40
	RNL-I—participation in social events (7–70)	54.3 (13.1)	8–70
**Frailty**	PRISMA-7 (0–7)	2.9 (1.4)	0–6
**Cognition**	MoCA (0–30)	21.2 (3.2)	16–28
**BADL**	FIM (18–126)	116.4 (9.2)	79–126
**IADL**	IADLq (0–23)	19.4 (3.7)	8–23

**Table 3 ijerph-19-01616-t003:** Correlation matrix between participation and the demographic, frailty, cognition, and ADL variables (N = 121).

	Participation		
	Total RNL-I	RNL-I Physical Activities	RNL-I Social Events
**Age**	−0.115	−0.084	−0.210 *
**Sex**	0.05	−0.021	0.112
**Cognition**	0.276 **	0.158	0.232 *
**Frailty**	−0.634 **	−0.657 **	−0.587 **
**BADL**	0.600 **	0.694 **	0.575 **
**IADL**	0.633 **	0.693 **	0.582 **

* *p* < 0.05, ** *p* < 0.001.

**Table 4 ijerph-19-01616-t004:** A linear multivariate regression analysis model to examine the contribution of frailty to explaining participation.

	Adjusted R^2^	Unstandardized B (SE)	Standardized Beta	Sig. F Change
Age	0.015	0.024 (0.416)	0.004	0.953
Cognition	0.08	0.379 (0.422)	0.068	0.371
Frailty	0.315	−5.21 (1.32)	−0.402	0.000
BADL	0.056	0.593 (0.178)	0.319	0.001

## Data Availability

The data presented in this study are available on request from the corresponding author.

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
