# Peer review of "The Contribution of Frailty to Participation of Older Adults"

_ijerph, 2022, doi:10.3390/ijerph19031616_

Round 1

Reviewer 1 Report

The paper addresses an important topic, however, the speculated hypothesis (that people with frailty experience more restrictions) is rather obvious. Therefore I recommend the authors to add more details on the nature of restrictions experienced, as this gives the reader information regarding caregiving needs etc.

  1. For the sentence "A negative moderate significant correlation was found between", please refer to the table in which the results are presented.
  2. It would be interesting to know how frailty characteristics are associated with restrictions. E.g., what aspects of frailty leads to most restrictions?
  3. It would also be interesting to know in what type of restrictions are experienced most/least. E.g., if you are frail, what are you still able to do? (make a phone call?) 
  4. Please add some thoughts on the clinical significance to the discussion.

Author Response

We thank the reviewers for their important comments.

Reviewer 2 Report

The manuscript discusses issues that, when understood properly, are essential for eldercare. The inclusion of participants of 75+ years of age is definitely a strength of the study (this subgroup can be reasonably expected to be less independent than commonly studied subjects - 60+ or 65+ years of age), and so is the selection of community-dwelling older subjects. The determinants of participation are particularly important in this group.

1. Introduction

Line 60: Therefore, frail older adults might experience difficulties in participating in activities that are meaningful to them, but this has not been studied.

This relationship has been studied, please revise this sentence (e.g., https://doi.org/10.1016/j.archger.2021.104616).

Line 69: Although gait in this recent systematic review was found to be an indicator of health and functioning, the relationship between reduced gait speed as one of the characteristics of frailty and participation has not been assessed.

Again, at least partial research results have been published in this area (e.g. https://doi.org/10.1016/j.pmedr.2016.06.005), please consider a less strict formulation.

2.6 Statistical analysis

Line 165: In order to control for age [13] and cognition (e.g. [37]), factors known to correlate with participation, they were entered first into the regression model.

This sentence reads imprecise, please streamline.

The results and conclusions are in line with expectations but there is value in investigating the kind and strength of relationships between analysed factors. The limitations of the study are presented appropriately, yet the size of the group must be considered relatively small (even if the effort to perform home visits is certainly quite high), given the distribution of analysed parameters.

It would also make sense to extend the characteristics of the study group (currently Table 2), grouping the participants in ranges related to presented parameters, in a manner similar to the sentence describing the cognitive status (page 5, no line numbers).

Author Response

(The authors gave the same response as above.)
